# Cortical mechanisms of across-ear speech integration investigated using functional near-infrared spectroscopy (fNIRS)

Gabriel G. Sobczak[1¤a]*, Xin Zhou[1¤b], Liberty E. Moore[1], Daniel M. Bolt[2], Ruth Y. Litovsky[1,3,4]

1 Waisman Center, University of Wisconsin–Madison, Madison, WI, United States of America, 2 Department of Educational Psychology, University of Wisconsin–Madison, Madison, WI, United States of America, 3 Department of Communication Sciences and Disorders, University of Wisconsin–Madison, Madison, WI, United States of America, 4 Department of Surgery, Division of Otolaryngology, University of Wisconsin–Madison, Madison, WI, United States of America

¤a Current address: Department of Otolaryngology, Indiana University School of Medicine, Indianapolis, IN, United States of America
¤b Current address: Brain and Mind Institute, The Chinese University of Hong Kong, Hong Kong SAR, China
* gsobczak@iu.edu

**Data Availability Statement:** Stimuli, unprocessed fNIRS data, and speech intelligibility data from this study are publicly available through the Open Science Framework. URL: https://osf.io/xdmwy/?view_only=2d17c98b9ae34ee9864b359c07bf0332

## Abstract

This study aimed to investigate integration of alternating speech, a stimulus which classically produces a V-shaped speech intelligibility function with minimum at 2–6 Hz in typical-hearing (TH) listeners. We further studied how degraded speech impacts intelligibility across alternating rates (2, 4, 8, and 32 Hz) using vocoded speech, either in the right ear or bilaterally, to simulate single-sided deafness with a cochlear implant (SSD-CI) and bilateral CIs (BiCI), respectively. To assess potential cortical signatures of across-ear integration, we recorded activity in the bilateral auditory cortices (AC) and dorsolateral prefrontal cortices (DLPFC) during the task using functional near-infrared spectroscopy (fNIRS). For speech intelligibility, the V-shaped function was reproduced only in the BiCI condition; TH (with ceiling scores) and SSD-CI conditions had significantly higher scores across all alternating rates compared to the BiCI condition. For fNIRS, the AC and DLPFC exhibited significantly different activity across alternating rates in the TH condition, with altered activity patterns in both regions in the SSD-CI and BiCI conditions. Our results suggest that degraded speech inputs in one or both ears impact across-ear integration and that different listening strategies were employed for speech integration manifested as differences in cortical activity across conditions.

## Introduction

The current study investigated across-ear integration for speech with both perceptual measures and application of functional near-infrared spectroscopy (fNIRS) to assess whether neural signatures of the across-ear integration are found at the cortical level. These phenomena were investigated under listening conditions that simulated cochlear implant (CI) processing

**Funding:** Funding for this study was derived from several sources, including National Institutes of Health – National Institute on Deafness and Other Communication Disorders (NIH-NIDCD) grant No. R01 DC003083 to Ruth Litovsky, the American Otological Society Fellowship Grant to Gabriel G. Sobczak, and in part by a core grant from the National Institute of Child Health and Human Development (P50 HD105353 to Waisman Center). The listed funders did NOT play a role in the study design, data collection and analysis, decision to publish, or preparation of the manuscript.

**Competing interests:** The authors have declared that no competing interests exist.

in one or both ears (peripheral degradation of speech information), compared to non-degraded listening conditions. The study was motivated by growing evidence that people with hearing loss who are fitted with prosthetic devices show deficits in being able to integrate information across the two ears. Of particular interest are CIs which are known to restore some functional hearing in listeners with profound hearing loss.

## Impaired across-ear integration in CI listeners

Listeners with bilateral CIs (BiCI), or with single-sided deafness (SSD) and CI in the deaf ear, hereafter referred to as SSD-CI, perform poorly compared to typical-hearing (TH) listeners when listening to speech in noisy environments. Successful speech perception in these scenarios depends on the binaural system, which facilitates separation of target speech from interfering sounds [1–4]. Impaired cortical-level integration due to poor peripheral speech encoding may be associated with these deficits, illustrated with reaction-time experiments in listeners with hearing loss [5]. Further, impaired auditory attention may compound the audibility issues associated with perceptual degradation in SSD-CI listeners [6]. An essential goal towards improving current aural rehabilitation strategies in listeners with profound hearing loss is identifying a neural signature indicative of across-ear integration in listeners with non-degraded and degraded signals in one or both ears.

## Alternating speech paradigm

In the present study, across-ear integration was examined using an alternating speech paradigm in which sequential segments of spoken sentences were alternated between ears at varying rates. This paradigm can produce a V-shaped speech intelligibility function whereby performance is best at low and high alternating rates, but poorer at middle rates [7]. These variations in performance across rates are likely manifestations of several perceptual phenomena that are involved in the process of integrating information from both ears. If sounds are presented to the two ears at the same time or at rapidly alternated between ears (>16 Hz), listeners can fuse information from the two ears to form a coherent auditory object [8–11]. When sounds are alternated slowly between ears (< 4 Hz), listeners might be able to switch attention between the ears to ultimately piece together content of running speech [7,12,13]. At intermediate rates, it may be the case that neither strategy is successful and speech understanding consequently declines.

Wesarg and colleagues used the alternating speech paradigm to examine binaural hearing in a study with SSD-CI listeners [14]. A V-shaped speech intelligibility function with minimum between 4 and 8 Hz was produced with monotic interrupted CI stimulus and with bilateral alternating stimulus, consistent with prior studies using alternating speech in TH listeners [7,12,13,15,16]. Binaural benefit (the difference between bilateral performance and monotic performance in the *hearing* ear) decreased monotonically from lower to higher alternating rates. The observed inverse relationship between binaural benefit and alternating rate across different listening configurations may indicate a decreasing reliance on both ears, and likely a reduction in attentional engagement to reconstruct speech as the alternating rate increases. In the present study, we combined behavioral and neuroimaging data (see below) in attempt to unravel the potential connection between utilization of different listening strategies across alternating rates and auditory attentional input associated with each listening strategy.

## Cortical regions of interest (ROIs)

Investigating the processing of alternating speech hinges both on the V-shaped speech intelligibility function and on auditory attention. Auditory cortex (AC) activity correlates strongly

with intelligible speech in fMRI studies [17,18]. Using fNIRS, Lawrence and colleagues similarly found that increased activity in the bilateral superior temporal regions correlated with higher intelligibility scores in TH listeners [19], and Olds found that activation in the superior temporal gyrus correlated with higher sentence accuracy scores in TH and CI listeners with good performance [20]. Together, these studies suggest that fNIRS measures of auditory cortical activity can provide a neural marker of speech intelligibility; therefore AC was chosen as our first *a priori* region of interest (ROI) when examining the processing of alternating speech.

The dorsolateral prefrontal cortex (DLPFC) is a component of the frontoparietal attention network involved in directing attention [21,22]. The DLPFC plays a role in attentional processing of sounds [23,24]. McLaughlin and colleagues found that left DLPFC activity, as recorded by combined electroencephalography (EEG) and magnetoencephalography (MEG), was related to acoustic feature task switches and further correlated with task performance [25]. These findings align with the recruitment of frontoparietal networks observed in visual or visual-spatial task switching [26,27], and suggest that the DLPFC is likely involved in both auditory/visual attention as well as attentional switching tasks. Therefore, DLPFC was chosen as another *a priori* ROI in the current study.

To the authors' knowledge, this is the first study to assess cortical activity in response to an alternating speech stimulus. However, it is important to note that fNIRS itself is not necessarily a novel functional neuroimaging modality and has been validated against fMRI data in multiple studies [28–31]. The decision to implement fNIRS, rather than other modalities such as fMRI or EEG, was largely based on the compatibility of fNIRS with the components of a CI, as listeners with CI are likely to be future participants in similar studies.

## Hypotheses and predictions

The present study investigated how spectral degradation affects across-ear integration of alternating speech. In one condition, non-degraded, typical hearing (TH) alternating speech was presented to both ears. SSD-CI and BiCI conditions were simulated in TH listeners by presenting vocoded speech in the right ear or both ears, respectively. We predicted that, if across-ear integration of speech differs across alternating rates in accordance with posited listening strategies, and addition of degraded input impairs across-ear integration, then speech intelligibility functions would vary across conditions, dependent both on alternating rates and on listening conditions. Specifically, speech intelligibility scores were predicted to be at ceiling across all rates in the TH condition, and to show canonical V-shaped function in the BiCI condition, with SSD-CI condition showing outcomes intermediated between the TH and BiCI conditions.

For fNIRS measures, differences in speech intelligibility scores were hypothesized to be associated with changes in cortical activity in the AC, with decreased AC activity under less intelligible listening conditions compared to more intelligible conditions. In conditions that showed V-shaped speech intelligibility across rates, particularly in the BiCI condition, we also predicted V-shaped trends in fNIRS responses in the AC. Higher DLPFC activity, reflective of auditory attention, would be observed in conditions that require greater attentional resources to reconstruct the speech content. Further, greater fNIRS responses in the DLPFC were predicted at lower alternating rates compared to higher rates, due to the posited switching of attention between ears at lower rates to reconstruct speech, and under conditions with degraded speech and alternating speech, i.e., BiCI and SSD-CI, compared to the TH condition.

## Methods

### Ethics statement

The protocol for this experimental study (protocol ID 2016-0226-CP022) was reviewed and approved by the University of Wisconsin-Madison Minimal Risk Institutional Review Board. Participants provided written, informed consent prior to the first data collection session.

### Participants

Twenty-four adult volunteers were recruited through a University of Wisconsin (UW) - Madison online job posting site between October 1st, 2020 and March 1st, 2021. One participant missed their scheduled sessions and did not return for testing. Two were excluded prior to data collection on account of poor data quality during calibration of the fNIRS system. Twenty-one participants advanced to the testing phase, and all who completed the study were paid for their time or received course credits. Data from one participant who completed the study were corrupted while uploading to a secure data storage server and the participant did not return to repeat the study. The final group of twenty participants (18 females and 2 males; age [mean ± standard deviation (SD)]: 20.3 ± 1.3 years, range: 18–23 years; 18 right-handed) were native monolingual English speakers with normal or corrected-to-normal vision, no known neurological disorders, and no significant musical experience. Pure tone thresholds were equal to or less than 20 dB HL with less than 10 dB difference between ears at octave frequencies between 250 and 8000 Hz. Experimental protocols followed National Institutes of Health standards and were approved by the UW - Madison's Human Subjects Institutional Review board.

### Stimuli

Speech stimuli consisted of sentences from the AuSTIN corpus [32], recorded by an American female speaker from our lab. These sentences contained three or four target words. An example sentence, with target words in bold, is "The room is very messy;" 640 unique sentences were selected for fNIRS testing, and a separate group of 120 sentences were selected for speech intelligibility testing. The distribution of sentences with three and four target words was kept constant across the two groupings. Both non-degraded and vocoded versions of the sentences were used. Vocoded sentences were produced with an 8-channel noise vocoder [33], consisting of a white-noise carrier and channels divided into eight frequency bands between 200 and 7000 Hz with filters based on Greenwood functions. Sentences were then modified in MatLab® (build R2020a) to alternate between ears at rates of 2, 4, 8, and 32 Hz; when speech was present in one ear, there was silence in the other (see Fig 1). Rates were selected to capture the V-shaped speech intelligibility function while minimizing the number of conditions required. For each of the alternating rates, three different speech conditions were used: non-degraded speech (TH), bilaterally vocoded speech (BiCI), and left ear non-degraded/right ear vocoded (SSD-CI). The audible periods of each sentence were normalized to the same overall root-mean-square (RMS) level.

Stimulus blocks for fNIRS sessions in each speech condition per alternating rate were generated by concatenating five sentences from the condition with an inter-stimulus interval of 500 milliseconds. After the sentence concatenation step, zero padding was implemented to each block both at the beginning and at the end to ensure a duration of 17 seconds. Twenty practice session blocks and 108 testing blocks were created. For the testing blocks, there were nine blocks for each of the twelve conditions (3 speech conditions x 4 alternating rates). All stimuli were delivered through ER-2A insert earphones (Etymotic® Research) and calibrated to be

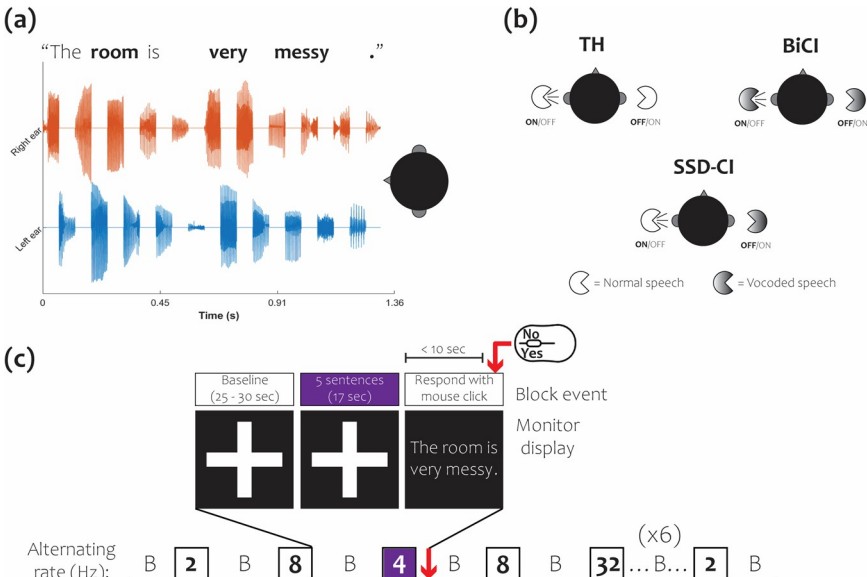

**Fig 1. Alternating speech stimulus, speech conditions, and fNIRS data collection.** (a) Spectrogram demonstrating an example non-degraded AuSTIN sentence, "The room is very messy," segmented and alternating between ears at 8 Hz. (b) Organization of speech conditions: NH = non-degraded speech in both ears; SSD-CI = vocoded speech in right ear, non-degraded speech in left ear; BiCI = vocoded speech bilaterally. (c) Pseudorandom block design, with stimuli in 4 listening conditions (boxes); 4 alternating rates presented at one speech condition in random order during each data collection session.

presented at 60 dBA ($F_{max}$, maximum level with A-weighted frequency response and fast time constant). For the SSD-CI speech condition, unprocessed speech in the left ear was attenuated by 3 dB to balance the subjective loudness of the unprocessed and vocoded speech [34–36].

## Assessment of speech intelligibility

The group of 20 participants were tested without fNIRS data collection to evaluate the effect of alternating rate and speech condition on speech intelligibility. Behavioral speech intelligibility testing was conducted without fNIRS recording because verbal responses to stimuli were required and would have likely introduced artefactual signals in the fNIRS data, primarily related to the movement of temporalis muscle during articulation. Since these artifacts would be tasked-evoked, they would have been challenging to exclude from the signals of interest [31,37–39]. Speech intelligibility testing occurred on days with four fNIRS testing sessions, following fNIRS data collection. Speech was presented through the insert earphones, and the experiment was run on a custom MatLab® script. Participants were instructed to listen to a single alternating AuSTIN sentence and repeat back as many words as they understood into a microphone. The experimenter then entered the number of correctly identified target words into the program. Sentences were played in groups of five at one speech condition (TH, BiCI, or SSD-CI) and one of four possible alternating rates. All four alternating rates were utilized before a new speech condition was played. The order of speech conditions and alternating rates was randomized across participants, and a total of 120 sentences were played (two groups of 5 sentences for each of the 12 conditions).

## Functional near-infrared spectroscopy (fNIRS) data collection

fNIRS is a non-invasive optical imaging modality ideally suited for auditory neuroscience experiments, given its temporal and spatial resolution, and compatibility with ferromagnetic and electrical components of hearing rehabilitation devices [20,40]. The present study utilized a continuous-wave near-infrared spectroscopy (NIRS) system with 16 light-emitting diode (LED) sources and 16 avalanche photodiode (APD) detectors (NIRScout[TM]; NIRX Medical Technologies, LLC). Each LED source emitted NIR light at 760 and 850 nm wavelengths. These wavelengths were used to evaluate the oxygenation state of hemoglobin (Hb), either oxygenated (HbO) or deoxygenated (HbR), in the brain tissue. APD detectors measured changes in the intensity of incident light, which were then converted to changes in HbO and HbR concentration using a modified Beer-Lambert equation. Neuronal activation is associated with increased regional cerebral blood flow, and consequently increased HbO and decreased HbR, a phenomenon known as neurovascular coupling [31,37]. The fNIRS response functions for HbO and HbR correlate well with the blood oxygenation level-dependent (BOLD) response in fMRI [41].

Each LED source with all adjacent APD detectors located at 30 mm distance constituted the measurement channels. In addition, eight detectors were placed 8 mm from their associated LED sources, creating "short channels" which recorded primarily extracerebral tissue responses [31,42]. The short-channel components were used to regress out the systemic noise in the regular channels and have been shown to significantly improve fNIRS signal-to-noise ratio [42]. Additional details about these short channels can be found in S1 File. The placement of light sources and detectors on the 10–10 system are detailed S1 Table. A NIRScap (NIRX Medical Technologies, LLC) fitted to participant's head circumference was used to arrange sources and detectors in the desired montage on their head (Fig 2A). fNIRS responses were examined in the DLPFC (**Fig 2**B) and AC (Fig 2C) on both hemispheres. Aggregate sensitivity profiles for each region were generated in the AtlasViewer program [43] with a generic "Colin27" brain anatomy atlas to ensure that our probe design yielded the highest probabilistic sensitivity to neuronal activity in the bilateral DLPFC and AC. The DLPFC was represented with Brodmann areas 45, 46, and 9; the AC was represented with Brodmann areas 41 and 42 [43].

All data were collected in a standard Industrial Acoustics Company sound-attenuated booth. A pseudo-random block design was implemented for fNIRS data collection and run in Presentation® [44]. Each participant underwent a total of nine 10-minute testing periods across two visits, with four or five testing periods per visit. The number of testing periods for visit 1 versus visit 2 was counterbalanced across participants. Each testing period began with a 30-second silent period for baseline data collection, followed by a stimulus block from one of the 12 conditions. A single testing period (Fig 1) contained twelve blocks from one speech condition (TH, BiCI, or SSD-CI) at randomized alternating rates (2, 4, 8, or 32 Hz) such that each alternating rate was repeated three times. The order of the nine testing periods was randomized across participants. In between each 17-second block, there was a silent period between 25–35 seconds in duration.

To ensure engagement during testing, participants were asked to attend to the 5 sentences in each block. Immediately after each block, a sentence was displayed on a monitor 1.5 m in front of the participant. Participants were asked to respond promptly as to whether the displayed sentence was played in the block or not, using a computer mouse button. In between blocks, participants were asked to focus on a white fixation cross displayed on the monitor. At the end of every 10-minute testing period, the participant's accuracy in identifying sentences was shown on the monitor as feedback. An abbreviated practice session was conducted prior to fNIRS testing to familiarize participants with the stimuli blocks and behavioral task at the

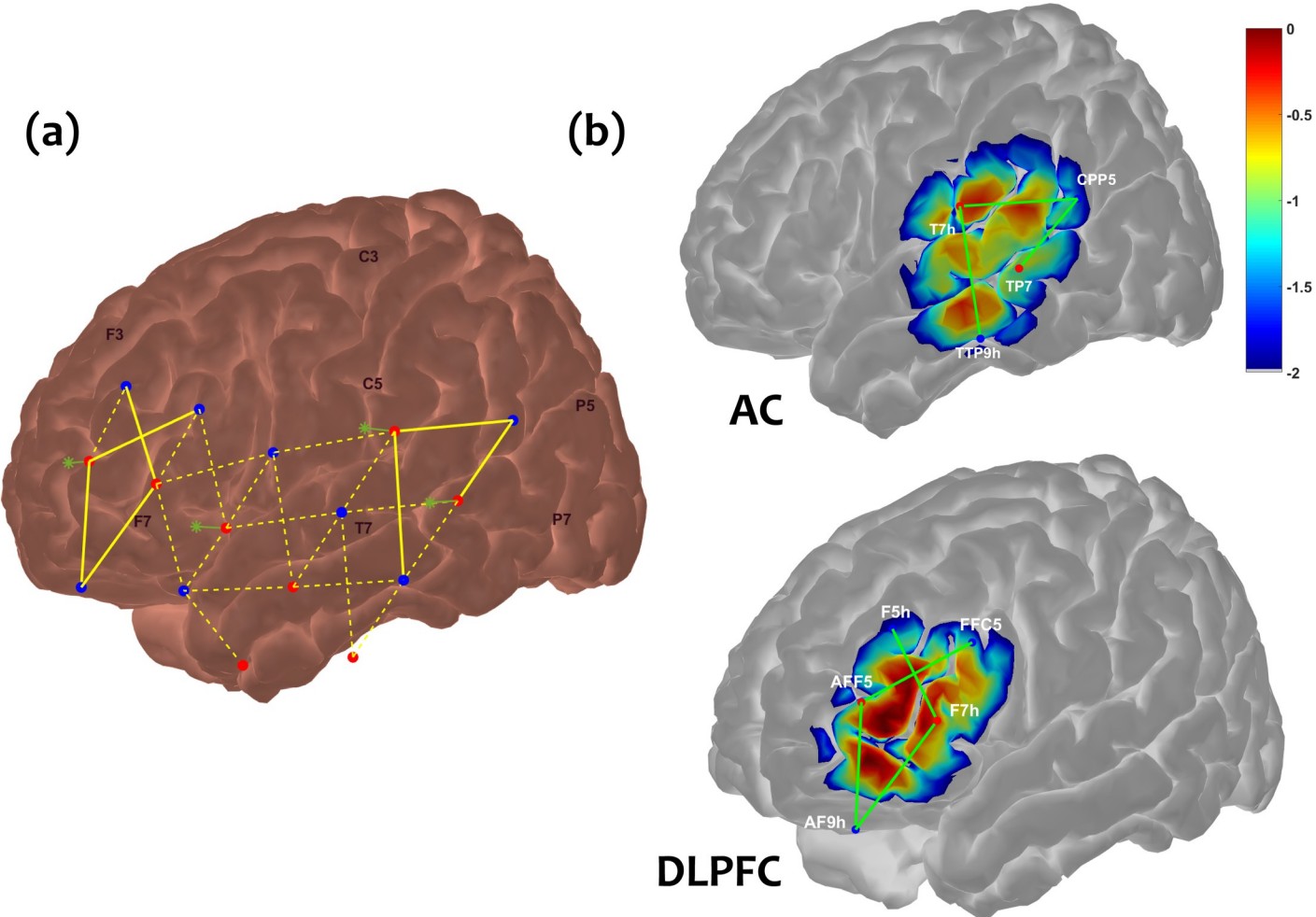

**Fig 2. Illustration of fNIRS montage and cortical regions of interest.** (a) fNIRS montage, showing sources (red dots), detectors (blue dots) and short-channel detectors (green asterisks). Dotted yellow lines connecting a source and detector represent measurement channels, while solid yellow lines represent measurement channels overlying cortical regions of interest (ROIs). fNIRS montage was symmetric between the left and right hemisphere. (b) Selected channels comprising a priori ROIs, the auditory cortex (AC) and dorsolateral prefrontal cortex (DLPFC); ROIs are only shown on the left hemisphere for demonstration purposes. Colormap corresponds to sensitivity profiles generated in AtlasViewer (43) for each ROI channel grouping, in units of $\log_{10}$ mm$^{-1}$.

beginning of each visit. There were 10 blocks per practice session, each block randomly selected from one of the 12 conditions with varying silent periods between blocks. Participants were asked to perform the same task as in the testing sessions. Verbal instructions were given by the experimenter, and text instructions were displayed on the monitor prior to testing. Accuracy on the sentence identification task was analyzed for button-push responses (true hits) in each speech condition across nine fNIRS recording sessions for each participant.

## fNIRS data analyses

fNIRS data were imported into MatLab® (build R2020a) for pre-processing and further analysis, using custom scripts written by the authors or scripts adapted from the Homer2 package [45]. The preprocessing pipeline consisted of: {1} rejecting channels of poorer quality (see S1 File and S2 Table for additional details), {2} converting light intensity to optical density, {3} reducing motion artifacts using a wavelet analysis method, {4} calculating the concentration changes in the hemoglobin, {5} filtering the low and high frequency noise

with a bandpass (0.01–1.5 Hz) filter, {6} reducing the noise from the extracerebral tissue using a general linear model (GLM) and principal component analysis (PCA) combined method (for details see [42] and [39]), and {7} applying another bandpass filter (0.01–0.09 Hz) to further remove respirations and heartbeat noise, then calculating the block-averaged responses for each ROI. A detailed description of wavelet analysis, GLM model, and block-average analysis is available in S1 File.

## Statistical analyses

The purposes of statistical analyses in the present study were as follows: {1} to examine the effects of alternating rate (2, 4, 8, and 32 Hz) and spectral degradation (by comparing SSD-CI and BiCI with TH) on speech intelligibility scores; {2} to examine the effects of alternating rate and spectral degradation on fNIRS responses, along with differences in responses between cortical regions of interest (AC, DLPFC) and brain hemispheres (left, right); {3} to elucidate potential relationships between fNIRS responses and speech intelligibility data. All analyses were performed in R (R Core Team, version 4.0.4, 2021).

For speech intelligibility, the percent-correct scores initially recorded for each participant were converted to rationalized arcsine units (RAU) to stabilize the variances of these scores [46,47]. The RAU scores were then tested for normality by running a Shapiro-Wilk test on the residuals from a two-way analysis of variance (ANOVA) with factors "alternating rate" and "speech condition". Because the residuals were not normally distributed ($W = 0.895$, $p = 6.88*10^{-12}$), an aligned rank transform (ART) analysis [48], which is a nonparametric analysis parallel to ANOVA, was conducted on the RAU scores. ART ("ARTool" R package) with a mixed model ("lmer" package) was performed on the RAU scores with alternating rate and speech condition as fixed factors and participant as a random factor. Post-hoc analyses within single factors were conducted using linear modeling ("lme4" package) and estimated marginal means ("emmeans" package) with Tukey method for $p$-value adjustment. When interactions between alternating rate and speech condition were tested post-hoc, the Holm method was used for $p$-value adjustment.

For the fNIRS measurements, ΔHbO and ΔHbR data were separately analyzed to assess the hemodynamic response in each ROI on the two hemispheres [49]. Normality of the ΔHbO and ΔHbR data were tested by running a Shapiro-Wilk test on the residuals of a four-way ANOVA with factors "alternating rate", "speech condition", "ROI", and "hemisphere". Neither the ΔHbO nor ΔHbR residuals were normally distributed ($W_{HbO} = 0.993$, $p_{HbO} = 1.23*10^{-4}$; $W_{HbR} = 0.993$, $p_{HbR} = 1.38*10^{-4}$). Hence, ART with mixed models were then separately performed on the transformed ΔHbO and ΔHbR data with alternating rate, speech condition, hemisphere, and ROI as fixed factors and participants as a random factor. Post-hoc analyses within single factors were conducted using linear modeling and estimated marginal means with Tukey method for $p$-value adjustment. No interaction analyses were performed as the omnibus ANOVA did not reveal any statistically significant factor interactions. Significant differences were found between both hemisphere and ROI factors, so post-hoc subgroup analyses were conducted on the four hemisphere-ROI combinations (left and right AC and DLPFC) to further examine the effects of alternating rate and speech condition on fNIRS responses. Because *a priori* hypotheses regarding subgroups were not generated, this analysis was exploratory in nature, in attempt to examine why no significant interactions were noted in the omnibus ANOVA. For the subgroup analysis, $p$ values were corrected for multiple comparisons using a false discovery rate method [50].

## Results

Analysis of experimental data is separated into behavioral speech intelligibility scores and functional neuroimaging data as recorded with fNIRS. For speech intelligibility, the canonical V-shaped intelligibility function was reproduced only in the BiCI condition. This finding suggests a potential effect of bilateral spectral degradation on intelligibility of alternating speech. Alternatively, there may be a unique contribution of the selected speech material, but that is an unlikely explanation for the canonical V-shaped function in the BiCI condition. TH and SSD-CI conditions had significantly higher scores and were not significantly different from one another across all alternating rates, and higher than the V-shaped reduction in scores in the BiCI condition. Sentence identification accuracy during the fNIRS experiment generally mirrored results from the behavioral experiment.

For fNIRS, the AC and DLPFC exhibited significantly different activity across alternating rates in the TH condition, with higher amplitude of ΔHbO in the DLPFC compared to the AC. Cortical activity was significantly impacted by the alternating rate of the speech stimulus, driven by increased ΔHbO amplitudes at 4 Hz compared with 8 Hz. However, the effect of alternating rate was not specific to any one ROI in subgroup analyses. Although activity patterns in both the DLPFC and AC were graphically altered in the SSD-CI and BiCI conditions compared to the TH condition, there was no significant main effect of speech condition on ΔHbO amplitude. Of note, in the left hemisphere, both the AC and DLPFC exhibited a V-shaped pattern of ΔHbO amplitudes, albeit with a minimum amplitude at 8 Hz instead of 4 Hz. ΔHbO and ΔHbR amplitudes were generally anticorrelated, which is expected for hemodynamic responses of neural origin. Further discussion of ΔHbR can be found in S1 File.

### Behavioral results–speech intelligibility experiment

Plotted in Fig 3 are participant-level RAU scores at each alternating rate and within-participant intelligibility functions. Group means and standard deviations across the three speech conditions are superimposed on each of the subplots. Speech intelligibility scores were generally lowest in the BiCI (bilaterally vocoded) condition and highest in the TH (bilaterally non-vocoded) condition, with SSD-CI (left non-vocoded, right vocoded) scores falling in between, in line with our predictions. The V-shaped speech intelligibility function was replicated in the BiCI condition but not in the SSD-CI condition, in contrast with our predictions.

Statistical analyses revealed a significant main effect of speech condition and alternating rate, and a significant interaction between speech condition and alternating rate (see the summary in Table 1). Post-hoc analyses revealed significant differences between BiCI and TH, and between BiCI and SSD-CI ($p < 0.001$). When comparing across alternating rates within one speech condition, the following contrasts were significantly different in the BiCI condition: 2–4 Hz, 2–8 Hz, 4–32 Hz, and 8–32 Hz ($p < 0.001$ for all). All contrasts were non-significant for the SSD-CI and TH speech conditions, indicating that there was not a significant change in speech intelligibility across alternating rates for these conditions, and suggesting "ceiling" performance in these two conditions. Interaction analyses are summarized in Table 1 and indicate that the effect of alternating rate was specific to the BiCI speech condition, since the V-shaped speech intelligibility function was only produced in this condition. Specifically, non-significant $p$ values were obtained between BiCI–TH and BiCI–SSD-CI at 2–32 Hz and 4–8 Hz, while remaining BiCI–TH and BiCI–SSD-CI contrasts were significant, indicate a V-shaped intelligibility function in the BiCI condition. The non-significant BiCI–SSD-CI at 8–32 Hz contrast illustrates that the average change in speech intelligibility scores between 8–32 Hz for both degraded speech conditions was similar (Fig 3). This finding neither refutes the V-shaped intelligibility function in the BiCI condition nor supports the presence of a V-shaped function

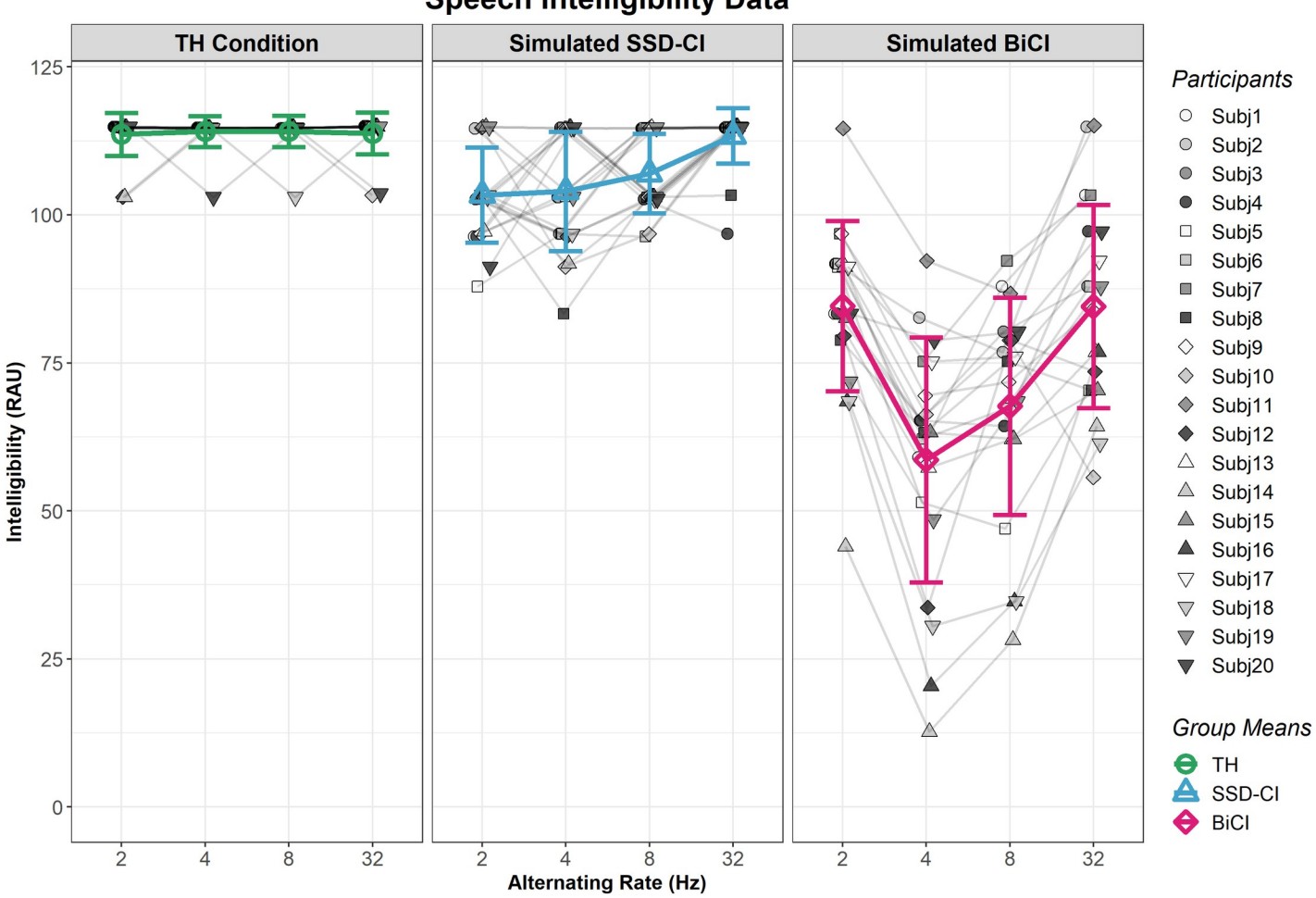

**Fig 3. Participant-level speech intelligibility data in rationalized arcsine units (RAU), at all four alternating rates.** Figure panels are separated by speech condition. Connecting lines between each participant's datapoints are added to illustrate participant-level patterns across alternating rates. Group-mean speech intelligibility ± 1 standard deviation (SD) is superimposed on participant-level data. Speech intelligibility data were collected in a separate behavioral experiment without fNIRS recording.

in the SSD-CI condition. Fifteen participants demonstrated a minimum at 4 Hz in the BiCI condition, while five participants (Subj2, Subj4, Subj5, Subj11, Subj 15) exhibited an 8 Hz minimum.

### Sentence identification accuracy during fNIRS recording

From the fNIRS sentence identification task, mean ± SD accuracy for each speech condition was as follows: TH– 95 ± 6%; SSD-CI– 94 ± 11%; BiCI– 89 ± 10%, indicating generally high accuracy on the task. There was a significant main effect of speech condition ($F(2,158) = 13.7$, $p < 0.001$), with significantly lower accuracy in the BiCI than in the TH and SSD-CI conditions ($p = 0.001$ and $p < 0.001$, respectively).

### fNIRS responses–DLPFC and AC

ΔHbO and ΔHbR data were analyzed in the *a priori* ROIs, i.e., the DLPFC and AC on both hemispheres. Fig 4 plots the group-mean block-averaged ΔHbO responses at each alternating

**Table 1. Summary of statistical results for speech intelligibility data.**

| | Mixed model analysis | Post-hoc | |
|---|---|---|---|
| Speech Cond | $F(2,209) = 328.42, p < 0.001$ | BiCI < TH; $p < 0.001$ BiCI < SSD-CI; $p < 0.001$ | |
| Alternating Rate | $F(3,209) = 23.49, p < 0.001$ | -- | |
| Speech Cond: Alternating Rate | $F(6,209) = 14.17, p < 0.001$ | -- | |
| *Contrasts between speech conditions and alternating rates* | | | |
| Contrasts | BiCI–TH ($p$) | BiCI–SSD-CI ($p$) | TH–SSD-CI ($p$) |
| 2–4 Hz | < 0.001 | < 0.001 | 1.00 |
| 2–8 Hz | < 0.001 | < 0.001 | 1.00 |
| 2–32 Hz | 1.00 | 0.20 | 0.21 |
| 4–8 Hz | 0.26 | 0.81 | 1.00 |
| 4–32 Hz | < 0.001 | 0.002 | 0.21 |
| 8–32 Hz | < 0.001 | 0.17 | 0.73 |

Bolded values indicate significance level of p < 0.01, after correction for multiple comparisons using Holm method.

rate, separated by speech condition, in these ROIs. Plots with standard error of the mean for AC and DLPFC traces are included in S1 and S2 **Figs**, respectively. Responses in the DLPFC generally had higher amplitudes than those in the AC. The right AC tended to have more consistent patterns of activation (positive amplitudes), whereas there appeared to be deactivation (negative amplitudes) in the left AC at 8 Hz in the SSD-CI (left non-degraded, right degraded) and BiCI (bilaterally degraded) conditions. Fig 5 plots the group means (bar graph) and standard error of the mean (error bars) of ΔHbO and ΔHbR responses in the same layout as Fig 4. In the left hemisphere, the AC and DLPFC responses in the TH (bilaterally non-degraded) condition exhibited opposite patterns, whereas in the SSD-CI and BiCI conditions, both the

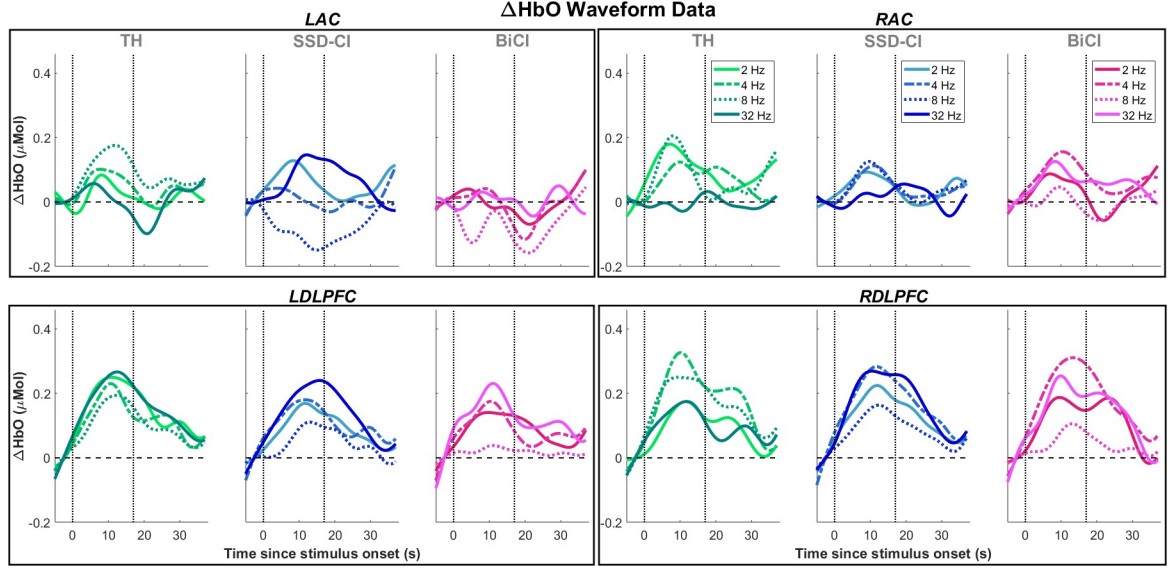

**Fig 4. Group-averaged ΔHbO waveforms in cortical regions of interest (ROIs).** Each plot contains waveforms at all four alternating rates. Columns correspond to the three speech conditions, and data from a single cortical ROI is contained in each black box. Within each plot, vertical dotted lines correspond to stimulus onset and offset.

AC and DLPFC exhibited minimum amplitude at 8 Hz. In the right hemisphere, at 4 Hz, the DLPFC showed maximal activity in all speech conditions, whereas AC showed maximal activity in the BiCI condition only.

Table 2 summarizes the statistical results for ΔHbO. Statistical results for ΔHbR are summarized in S3 Table. There were significant main effects of alternating rate ($p = 0.03$), ROI ($p < 0.001$), and hemisphere ($p = 0.01$), with a non-significant effect of speech condition ($p = 0.09$). There were no significant interactions between any of the fixed factors. *Post hoc* analyses revealed that DLPFC response amplitudes were higher than AC amplitudes, and that right hemisphere amplitudes were higher than in the left hemisphere. The main effect of alternating rate was driven by significantly higher response amplitudes at 4 Hz than at 8 Hz ($p = 0.04$), and marginally lower amplitudes at 8 Hz than at 32 Hz ($p = 0.07$). Given the negative ΔHbO at 8 Hz in the LAC for SSD-CI and BiCI conditions (see Fig 5), it is possible that this effect was moderated both by increased ΔHbO at 4 Hz and *decreased* (more negative) ΔHbO at 8 Hz. However, our statistical analysis was not adequately powered to assess this level of detail. These findings may indicate changes in cortical activity due to implementation of new listening strategies as the alternating rate changed.

Subgroup analyses were employed to evaluate the effects of alternating rate and speech condition in each of the four ROIs. This analysis was exploratory in nature. Results from this analysis are tabulated in Table 2. The main effect of speech condition with ΔHbO data in the left AC was non-significant ($F(2,893) = 2.30$, $p = 0.10$), and the main effect of alternating rate with ΔHbO data in the left and right DLPFC similarly non-significant ($F(3,893) = 2.20$, $p = 0.09$; $F(3,893) = 2.24$, $p = 0.08$, respectively). As the estimated effects are of moderate magnitude, the

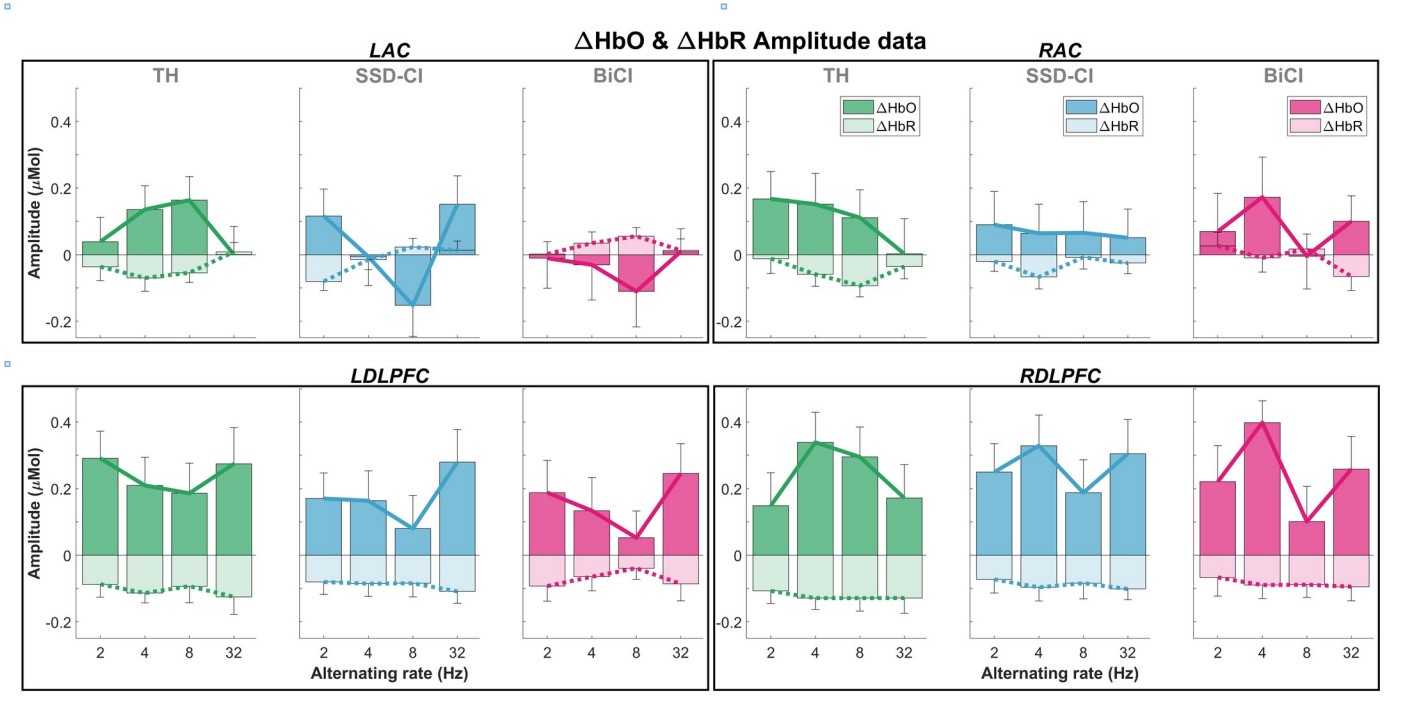

**Fig 5. Group-averaged ΔHbO and ΔHbR amplitude data, in the cortical regions of interest (ROIs).** Amplitudes are plotted across alternating rates, and error bars correspond to ± 1 standard error of the mean (SEM). Connecting lines between amplitude bars (solid = ΔHbO, dotted = ΔHbR) demonstrate overall patterns in amplitude changes across alternating rates. Columns contain data from one speech condition, and black boxes contain data from one cortical ROI.

**Table 2. Summary of statistical results for ΔHbO amplitudes.**

| Mixed model analysis | | post-hoc | |
|---|---|---|---|
| Speech Cond | $F(2,893) = 2.37$, $p = 0.09$ | | |
| Alternating Rate | $F(3,893) = 2.97$, $p = 0.03$ | 4 Hz > 8 Hz; p = 0.04 8 Hz < 32 Hz; p = 0.07 | |
| ROI | $F(1,893) = 43.52$, $p < 0.001$ | DLPFC > AC, $p < 0.001$ | |
| Hemisphere | $F(1,893) = 6.86$, $p = 0.01$ | Right > Left, $p = 0.01$ | |
| post-hoc subgroup analysis | | | |
| Brain Region | Left AC | Right AC | Left DLPFC | Right DLPFC |
| Factor | $p$ | $p$ | $p$ | $p$ |
| Speech Cond | 0.10 | 0.83 | 0.19 | 0.12 |
| Alternating Rate | 0.32 | 0.54 | 0.09 | 0.83 |
| Speech Cond: Alternating Rate | 0.16 | 0.62 | 0.96 | 1.00 |

lack of statistical significance is likely impacted by the smaller sample size in the subgroup analyses.

For the above exploratory subgroup analyses, the authors note that differences in the statistical significance of effects observed across subgroups does *not* imply presence of an interaction; no claims are made about differential subgroup effects with this analysis.

## Discussion

In this study we measured perception of speech that alternated between the two ears at different rates, and fNIRS measures of cortical activation patterns in response to the same stimulus. Of interest was a comparison across conditions between bilaterally non-vocoded (TH) and vocoded speech in the right ear only (simulated SSD-CI) and bilaterally vocoded (simulated BiCI). A V-shaped intelligibility function was reproduced in the BiCI speech condition, with a significant decrement in intelligibility at all rates compared to the TH and SSD-CI conditions, but no significant difference across alternating rates between the latter two conditions. fNIRS measures revealed a significant effect of the 4–8 Hz alternating rate contrast on ΔHbO in all four regions, with higher HbO amplitudes in the DLPFC compared to the AC, and in the right hemisphere compared to the left.

### Patterns in behavioral speech intelligibility data and sentence identification

Our study found similar behavioral results to previous work with alternating speech, as published by Wesarg (14) indicating replicability of data–the study utilized the German OlKiSa corpus (Oldenburg Sentence Test for Children) which contains sentences similar in structure to those of AuSTIN. Of note, fewer alternating rates were used in the current study compared to these prior experiments, focusing on the end point and middle points, aiming to reduce experiment duration and burden on participants. Speech intelligibility scores were lowest (worst) in the BiCI condition among the three conditions. This was an expected finding, in keeping with psychoacoustic research which demonstrates that increasingly degraded speech (in the present study, bilateral vocoded speech) correlates with progressively reduced intelligibility [e.g. [51–54]]. Additionally, in the BiCI condition, a V-shaped intelligibility function was obtained, suggesting different strategies were implemented across alternating rates. Work from [55,56] may explain the observed point of minimum speech intelligibility with

alternating speech stimuli. Both groups studied the effects of sinusoidally modulating the inter-aural phase difference of interfering noise presented with diotic speech. Speech intelligibility decreased with modulation frequencies above 4–5 Hz, suggesting that the noise became fused as a single percept and interfered with target speech at this critical modulation rate. Therefore, the 4 Hz speech alternation rate potentially represents a boundary where listeners no longer selectively attending to each ear and where binaural fusion occurs for stimuli that alternate across the two ears.

The finding of ceiling-level scores in speech intelligibility in the TH (non-vocoded) and SSD-CI (left non-vocoded and right vocoded) conditions were in contrast with results in previous studies, but unsurprising. We propose that whether data appear to assume a "V" shape function is influenced by the duration of speech material, a proxy for cognitive load [57–59]. Previous studies examining intelligibility to alternating speech utilized a shadowing paradigm [7,13,15,16,47], whereby listeners repeated back in real time a spoken passage of 100–125 words alternating between ears. These studies obtained a point of minimum intelligibility at 4–6 Hz using non-degraded speech in TH listeners. In contrast, when speech materials were short and the tasks were simpler, the V-shaped function was not found. Wesarg [14] utilized the German Oldenburg Sentence Test for Children [60], in which sentences are pre-set to a 5-word length, similar to the AuSTIN corpus in the current study. In line with our results, the intelligibility function for real SSD-CI listeners in that study had a shallow U-shape: there were no significant differences in speech intelligibility across intermediate alternating rates of 2.6, 4, and 8 Hz.

To that end, we propose that both the presence of degraded speech input and the duration of speech material may influence the shape of the intelligibility function. However, a connection between "cognitive load" and speech intelligibility may not be so straightforward [61]. Arguing against our proposal, Stewart and colleagues performed an experiment where shadowed passages were paused for participants' responses after clauses and sentence boundaries, resulting in an average segment size of 8.5 words [16]. With this modification, participants performed better compared to when shadowing the passages, but the V-shaped intelligibility function was still maintained across alternating rates. More research using within-participant comparisons across different sentence corpuses is required to address this hypothesis.

We considered potential effects of exposure to AuSTIN sentence structure on the speech intelligibility scores. Note that the behavioral speech intelligibility experiment was run after several fNIRS testing sessions, or interval between two visits, and the same AuSTIN sentence structure was used in both speech intelligibility and fNIRS testing sessions [62,63]. However, our results, detailed in S1 File, found that there was no learning effect of Day 1 vs. Day 2 testing on speech performance (p = 0.55), or test-retest interval on Day 2 performance (p = 0.66). If participants had become more familiar with stimuli over multiple sessions, it was not sufficient to allow them to predict speech content, or they did not attempt to predict speech content despite familiarity with the stimuli.

## Alternating speech and markers of speech intelligibility in the AC

Responses in the auditory cortex (AC) were predicted to mirror speech intelligibility scores measured separately in the behavioral speech perception session, based on prior neuroimaging studies that found direct correlations between AC activity and speech intelligibility [17–20] and fNIRS data collected in our lab [64]. However, this pattern was not observed in the TH (non-vocoded) speech condition. The left AC showed an inverse V-shaped trend in ΔHbO amplitude arose across alternating rates in the SSD-CI (left non-vocoded and right vocoded) and BiCI (bilaterally vocoded) conditions, whereas the right AC generally showed a monotonic

decrease in activity as alternating rate increased. When vocoded speech was present in the SSD-CI and BiCI conditions, cortical activation patterns in the AC changed when compared to the TH condition. In the left AC, there was a marked V-shaped pattern in ΔHbO amplitude that emerged in both the SSD-CI and BiCI condition, contrasting the inverse V-shaped trend in the TH condition. The left AC encodes intelligibility of speech–especially sentences [65–68], whereas the right AC encodes features of speech such as the envelope [69,70]. These findings may account for why the left AC responses in SSD-CI and BiCI conditions closely mirrored the shape of speech intelligibility functions across alternating rates in our experiment.

The presence of the 8 Hz minimum in both SSD-CI and BiCI speech conditions in the left AC may indicate a neural marker of intelligibility for degraded, dichotic speech stimuli. Behaviorally, there was a graphical minimum for intelligibility at 4 Hz in the BiCI condition, but there were no significant differences in RAU scores between 4 Hz and 8 Hz. However, five participants exhibited 8 Hz minima speech intelligibility in the BiCI condition and may have driven the average fNIRS response towards a minimum at 8 Hz rather than 4 Hz. The fact that this negative ΔHbO amplitude is isolated to the left AC is potentially related to left hemispheric dominance of temporal processing of amplitude-modulated speech [71–73], as our alternating stimulus was sequentially modulated by zeros and ones. This left dominance is typically limited to short temporal windows ($< 100$ ms), and perhaps is most salient for correspondingly higher alternating rates. The authors propose that the shape of the fNIRS responses in the left AC should be viewed more broadly in the context of the V-shaped speech intelligibility function, and that whether the ΔHbO response amplitudes are positive or negative at a given alternating rate does not necessarily have a physiologic correlate. It should be noted that our analyses of fNIRS data could not parse out differences across alternating rates in the BiCI condition, at the level of the left AC. Further research into the observed graphical discrepancy between behavioral and neuroimaging data is required.

The ΔHbO data demonstrated a statistically significant main effect of hemisphere, with higher-amplitude responses in the right compared to left hemisphere. This could in part be accounted for by the observation that the right AC is sensitive to degradation of speech [74–77] and right AC activation is enhanced by attentive listening [78–81]. Examining the right AC, a new pattern of ΔHbO responses emerged in the BiCI condition when compared with the TH condition. Interestingly, there was a maximum at 4 Hz, aligning with right DLPFC activity and opposing the predicted minimum response activity at 4 Hz. It is possible, then, that the right AC is more sensitive specifically to degraded speech stimuli compared to the left AC. However, interpreting the laterality of auditory alternating speech processing is challenging. Alternating speech is most appropriately categorized as a dichotic speech stimulus (14), or an instance when a different stimulus is present at each ear. Dichotic stimuli can be represented in either the left or right AC, depending on which hemisphere is contralateral to the target stimulus [82–84]. The dichotic stimuli in this study could also be construed as connected speech stimuli because the task necessitated that listeners reconstruct the stimulus into whole sentences. Accordingly, one might expect left-dominant AC representation [85], but right hemispheric dominance (in terms of higher response amplitudes) was shown in our data; this may indicate a yet unresolved processing of alternating, degraded speech. To our knowledge, no prior neuroimaging studies involving a similar alternating speech stimulus have been conducted; further research is required to elucidate the hemispheric lateralization in alternating speech processing with and without degradation of the stimulus.

## The role of the DLPFC and auditory attention in processing alternating speech

Distribution of auditory attention likely plays a key role in across-ear integration, especially in SSD-CI and BiCI listeners where the inputs are not synchronized across ears as in TH listeners [86,87]. "Auditory attention" refers to the top-down, voluntary process of focusing cortical processing resources to enhance behavioral sensitivity towards, and informational processing of, auditory stimuli [21]. In the right DLPFC, increased response amplitudes were noted at 4 and 8 Hz in the TH condition in the current study, in line with our expectation for increased frontal activation at intermediate alternating rates. In the presence of vocoded speech, this response pattern seemed to be conserved, with peak activation at 4 Hz in both the SSD-CI and BiCI condition. The significance of the biphasic appearance with a second peak at 32 Hz is unclear, despite speech intelligibility being essentially equivalent at 2 Hz and 32 Hz. The right DLPFC is thought to be involved in auditory spatial selective attention [88–90]. If listeners are switching attention between ears at certain rates, the alternating speech stimulus could be considered a spatial selective attention task, supporting the observed preferential right hemisphere activation.

The left DLPFC showed opposing activation patterns to the right DLPFC, with decreased activity at 4 and 8 Hz alternating rates in the former, which could be a manifestation of hemispheric differences in attentional processes. Post hoc analyses showed that the right hemisphere had higher response amplitudes than in the left. The left DLPFC activity may reflect attentional switching rather than direction of spatial attention [91–93], and these separate cortical functions of the left and right DLPFC may account for the "out-of-phase" behavior observed in the TH speech condition. Further, the observation that overall response patterns were consistent across speech conditions within the left and right DLPFC suggests conserved, but disparate, functions in attentional processing of alternating speech segments.

Alternatively, opposing activation patterns between the right and left DLPFC could represent hemispheric differences in degraded speech processing [39,94–98]. The left DLPFC exhibited a point of minimum activation at 8 Hz in the presence of vocoded speech in one or both ears, corresponding to the point of minimal activation at 8 Hz in the left AC. Whereas the right DLPFC only appeared to mirror right AC activity in the BiCI condition, in which there was a pronounced peak activation at 4 Hz that graphically corresponds with the 4 Hz speech intelligibility minimum point.

## Evidence of cortical-level, across-ear speech integration

Based on binaural benefit data obtained using alternating speech stimuli in different listening conditions [14,99], it is evident that listeners rely on both ears at low alternating rates, and on one ear at high alternating rates, to reconstruct the alternating sentences. This observation indicates potential utilization of different listening strategies depending on the alternating rate: switching attention between ears at low rates and focusing on one ear at high rates. fNIRS data revealed that differences between 4 and 8 Hz drove the main effect of alternating rate in ΔHbO responses, implying a global shift in cortical activity between these two alternating rates. These results support inferences from behavioral data that different listening strategies governed how participants reconstruct the alternating speech stimulus. The potential fNIRS evidence for binaural fusion of the alternating speech at a critical alternating rate (4–8 Hz), and how this neural signature of fusion is impacted by peripheral degradation of the alternating speech (vocoding), merits discussion.

Our fNIRS data revealed a significant main effect of alternating rate in ΔHbO responses, driven by the 4–8 Hz contrast. This finding potentially indicates a change in cortical activity at

intermediate alternating rates, which could align with a shift in listening strategy from switching attention between ears as the alternating rate increased. However, it may also reflect distinct neural processing of vocoded speech occurring in the left AC at 8 Hz alternating rate, given the strong negative ΔHbO in the SSD-CI and BiCI conditions at this rate. The authors favor the former explanation; however, our data is unable to entirely exclude the latter explanation. There was also a main effect of ROI in the ΔHbO responses, indicating that the AC and DLPFC contributed differently to the processing of alternating speech. However, predicted response patterns were not recapitulated in the four regions examined. Given the unique speech stimulus that necessitated across-ear integration to reconstruct the sentence segments, it is possible that the lack of neat alignment between predicted and observed fNIRS responses in the four brain regions may indicate a component of binaural processing at the cortical level.

Regarding the impact of degraded speech on binaural processing, prior research has demonstrated that early asymmetric hearing loss, either temporary or more permanent, impacts auditory cortical representation of binaural stimuli and that cochlear implantation only partially restores the representations observed in TH listeners [100–104]. Assuming our fNIRS results do capture a component of binaural processing at the cortical level, then we have further demonstrated that peripheral degradation of speech impacts cortical-level across-ear integration. Our fNIRS data showed a significant main effect of speech condition in the ΔHbR data; this effect did not reach significance in the ΔHbO data. Both ΔHbR and ΔHbO correlate with task-evoked cerebral hemodynamic changes [105–107] and are generally anticorrelated [31,108]. However, ΔHbO is favored in the literature due to higher absolute and relative response magnitudes and stronger correlations with task-evoked responses [109]. Some experiments have shown ΔHbR correlation with task-evoked responses [110], but the relevance of significant ΔHbR in absence of significant ΔHbO is less studied. Hence, the fNIRS data from the present study might support previous observations that spectral degradation of auditory inputs impacts cortical signatures of across-ear integration, assuming that significant ΔHbR are physiologically relevant.

## Limitations

There are several limitations in this study. First, our sample size was likely not adequate to fully reveal effect sizes in the fNIRS data. A power analysis was not conducted because there have not been previous fNIRS or other neuroimaging studies examining effects like those explored in the present study. Because it was an in-person study run during the early stages of the COVID-19 pandemic, participant recruitment was challenging, impacting the sample size. We have reported some non-significant statistical findings as marginally non-significant or analogously. Potential trends illuminated by these findings may be of interest to our readers given the novelty of the data despite the lack of power. The "marginally non-significant" statistical results must be interpreted cautiously given our analytical approaches.

Second, at the time that this study was conducted, due to Covid-19 restrictions placed on research protocols, the duration of in-person research studies was limited to 90 minutes per session, thus, conditions were selected to best recapitulate the V-shaped speech intelligibility function while minimizing experiment duration. Previous work has shown that the 32 Hz alternating rate produces ceiling intelligibility performance [7,14]. Neither a bilateral nor a monaural non-segmented (control) speech condition was included, nor was a monaural segmented condition included with which binaural benefit could be calculated. Without a monaural control or behavioral measures of across-ear integration, interpretation of the neuroimaging findings as evidence of impaired across-ear integration may be somewhat limited. Numerous studies have established different patterns of cortical dynamics in response to

monaurally versus binaurally presented signals [39,111–115]. Furthermore, studies in bimodal cochlear implant listeners (those with a cochlear implant and contralateral hearing aid) have demonstrated that bimodal (binaural) benefit, which is calculated with data from monoaural and binaural conditions, can be correlated with changes in auditory evoked potentials [116,117]. Hence, without monaural control data, we generally rely on the body of existing literature to support our inferences that the behavioral and fNIRS results from our study are a manifestation of across-ear integration.

Data analysis was limited by the variability present in the fNIRS data. Differences in cap positioning between participants may have caused data to be recorded from slightly different cortical regions across participants, contributing to the variability. Wijayasiri and colleagues measured the inter-participant variability in cap placement with their fNIRS study, finding a difference of 6.64 ± 0.53 mm (mean ± SD) between participants (98), which the authors considered adequate given the 30 mm channel spacing. An analogous procedure of cap positioning was utilized in the present study, so cap placement probably contributed minimally to fNIRS data variability. fNIRS data is known to be intrinsically variable between participants and group averaging over many participants is necessary to reveal underlying neural dynamics [31,37,118]. This impacted statistical results: because the data was not normally distributed, outliers could not be excluded using parametric tools such as Grubb's [119] test. We attempted a data-driven outlier exclusion by searching for poor performers in the speech intelligibility test, as poor behavioral test performance predicts anomalous brain activation patterns because the participant is not actively engaged in the task [120–122]. This did not prove to be a fruitful methodology, as participants may have had one or two outlying behavioral data points, but none were global poor performers across alternating rates within one speech condition, or at one alternating rate across all speech conditions. Hence, excluding the participant's fNIRS data would not have been valid. As a result, many of our $p$ values in post-hoc tests were marginally non-significant, and some significant findings became non-significant after correction for multiple comparisons. The rationale for still reporting these findings is that this study was in part exploratory, and they may be of interest to readers planning similar studies. Nonetheless, interpretation of these results must be approached very cautiously.

Lastly, limitations of the alternating speech stimulus itself should be discussed. A similar stimulus, the Rapid Alternating Speech Perception (RASP) test [123], was posited to assess binaural fusion at the level of the auditory brainstem and was included in the clinical testing battery for central auditory processing disorders (APD). As early as the 1980s, the validity of RASP was called into question. Shea and Raffin recommended that RASP not be used due to a lack of established "normal" test results [124]. Harris and colleagues performed a study in which 24 participants listened to one channel or the other of RASP (i.e., monotic interrupted speech) [125]. Mean sentence scores were 37.7% and 20.8% for channel 1 and channel 2, respectively, and it was concluded that RASP was not an ideal test for binaural fusion because a single channel should contain minimal intelligible speech information. A group of licensed audiologists commonly used RASP and masking level difference (MLD) to assess binaural fusion, but less than 25% of survey respondents reported implementing RASP in their practice [126]. Current audiology practice utilizes acoustic reflexes, auditory brainstem responses (ABRs), and MLDs to assess the function of the low auditory brainstem. RASP tends to be avoided due to its lack of correlation with ABRs [127]; however, mixed evidence exists for correlation of MLDs with ABRs [127–129]. Despite the drawbacks of RASP as a clinical audiologic instrument, alternating speech is nonetheless proposed as a valid stimulus to investigate differences in speech integration across the ears at cortical levels of the central auditory pathway.

## Conclusions

The present study investigated how spectral degradation affects across-ear integration of alternating speech and whether fNIRS could reveal across-ear integration of speech at cortical level. Our behavioral results replicate and add to findings in the literature: alternating speech produces a V-shaped intelligibility function with minimum at 4 Hz alternating rate when speech is bilaterally vocoded that simulated hearing in BiCI listeners. Our fNIRS data, novel in the literature, reveals that {1} the AC and DLPFC are differentially involved in processing alternating speech with a specific impact of intermediate alternating rates, and that {2} response patterns are changed by the presence of degraded speech in one or both ears compared to the TH condition. The objective (fNIRS) data complements behavioral data that it may reveal differences in auditory stimulus processing strategies not necessarily evident in behavioral data alone and reflects a dominant listening strategy of across-ear speech integration at 4–8 Hz. We hope that this study will provide a foundation to better understand observed binaural hearing deficits in cochlear implant listeners compared to their TH listeners counterparts, and in the future inform individualized auditory rehabilitation strategies to provide CI listeners with maximal functional benefit.

## Supporting information

**S1 Fig. Group-averaged ΔHbO waveforms in bilateral AC, with standard error of mean response amplitude.** Each plot contains waveforms at one alternating rate for one speech condition. Linear traces indicate mean response amplitude, and shaded regions correspond to the standard error of the mean response amplitude. Columns correspond to the three speech conditions, and data from a single cortical ROI is contained under the solid horizontal line with corresponding ROI label. Within each plot, vertical dotted lines correspond to stimulus onset and offset.
(TIF)

**S2 Fig. Group-averaged ΔHbO waveforms in bilateral DLPFC, with standard error of mean response amplitude.** Each plot contains waveforms at one alternating rate for one speech condition. Linear traces indicate mean response amplitude, and shaded regions correspond to the standard error of the mean response amplitude. Columns correspond to the three speech conditions, and data from a single cortical ROI is contained under the solid horizontal line with corresponding ROI label. Within each plot, vertical dotted lines correspond to stimulus onset and offset.
(TIF)

**S1 Table. Placement of fNIRS sources (S, n = 16) and detector (D, n = 16) on the 10–10 system.**
(TIF)

**S2 Table. Participants with 1 remaining measurement channel in either auditory cortex following channel exclusion protocols.**
(TIF)

**S3 Table. Summary of statistical results for ΔHbR amplitudes.**
(TIF)

**S1 File. Supplementary materials.** This is the document containing additional information pertinent to the study for readers to reference. **Open Science Framework file-sharing link:** https://osf.io/xdmwy/?view_only=2d17c98b9ae34ee9864b359c07bf0332.
(DOCX)

## Acknowledgments

The authors appreciate the time, support, and willingness of all participants. We also thank our colleagues from the Binaural Hearing and Speech Lab who assisted with participant recruitment and for suggestions regarding experimental design and implementation, including Shelly Godar and Z. Ellen Peng.

**Author's note**

Portions of the data were presented at the Association for Research in Otolaryngology Virtual Midwinter Meetings (February 2021 and 2022), UW-Madison Division of Otolaryngology Virtual Resident Research Day (June 2021), virtual Conference on Implantable Auditory Prostheses (July 2021), and American Academy of Otolaryngology–Head and Neck Surgery Annual Meeting (October 2021). Stimuli and data from this study are publicly available through the Open Science Framework. The file-sharing link is included here and in Supporting Information.

## Author Contributions

**Conceptualization:** Gabriel G. Sobczak, Xin Zhou, Ruth Y. Litovsky.

**Data curation:** Gabriel G. Sobczak, Xin Zhou.

**Formal analysis:** Gabriel G. Sobczak, Xin Zhou, Daniel M. Bolt.

**Funding acquisition:** Gabriel G. Sobczak.

**Investigation:** Gabriel G. Sobczak, Liberty E. Moore.

**Methodology:** Gabriel G. Sobczak, Xin Zhou, Liberty E. Moore, Ruth Y. Litovsky.

**Project administration:** Gabriel G. Sobczak, Ruth Y. Litovsky.

**Resources:** Xin Zhou, Ruth Y. Litovsky.

**Software:** Xin Zhou, Liberty E. Moore.

**Supervision:** Xin Zhou, Ruth Y. Litovsky.

**Validation:** Liberty E. Moore.

**Writing – original draft:** Gabriel G. Sobczak.

**Writing – review & editing:** Gabriel G. Sobczak, Xin Zhou, Liberty E. Moore, Daniel M. Bolt, Ruth Y. Litovsky.

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
