## [Decision Letter · Decision Letter 0]

4 Mar 2024

PONE-D-23-42460Cortical Mechanisms of Across-Ear Speech Integration Investigated Using Functional Near-Infrared Spectroscopy (fNIRS)PLOS ONE

Dear Dr. Sobczak,

Thank you for submitting your manuscript to PLOS ONE. After careful consideration, we feel that it has merit but does not fully meet PLOS ONE’s publication criteria as it currently stands. Therefore, we invite you to submit a revised version of the manuscript that addresses the points raised during the review process.

**ACADEMIC EDITOR: **

The paper can be revised upon reviewers' comments. Please revise and submit the revised version. 

We look forward to receiving your revised manuscript.

Kind regards,

Noman Naseer, PhD

Academic Editor

PLOS ONE

Journal Requirements:

2. Please expand the acronym “NIH-NIDCD” (as indicated in your financial disclosure) so that it states the name of your funders in full.

Additional Editor Comments:

The paper can be revised upon reviewers' comments. Please revise and submit the revised version.

Reviewers' comments:

Reviewer's Responses to Questions

**Comments to the Author**

1. Is the manuscript technically sound, and do the data support the conclusions?

Reviewer #1: Partly

Reviewer #2: Yes

2. Has the statistical analysis been performed appropriately and rigorously? 

Reviewer #1: Yes

Reviewer #2: Yes

3. Have the authors made all data underlying the findings in their manuscript fully available?

Reviewer #1: Yes

Reviewer #2: Yes

4. Is the manuscript presented in an intelligible fashion and written in standard English?

Reviewer #1: Yes

Reviewer #2: Yes

5. Review Comments to the Author

Reviewer #1: The manuscript investigates and assess potential cortical signatures of across-ear integration of alternating speech. The study utilizes fNIRS technique to record brain signals and after processing found similar results to previous work with alternating speech. The manuscript is well-structured and has performed fNIRS data acquisition and analysis. However, it briefly mentions limitations, such as potential effects of exposure to sentence structures and the need for further research on the observed graphical discrepancy between behavioral and neuroimaging data.

Some suggestions are there to increase the effectivity of this research as

1. Which type of noise was removed by applying bandpass of 0.01–0.5 Hz? Why choose only this band?

2. The results were discussed very briefly and description of results may help the reader to better understand.

3. In statistical results for speech intelligibility, multiple p–values are non-significant, that should be discussed in the results as well and need explanation.

4. The study claims fNIRS data acquisition and analysis as novel contribution, the comparison of fMRI and EEG for the same experiment should be included to strengthen the claim and also to investigate what significance contribution fNIRS brought to the study.

5. Ethical statement and IRB approval reference should be added in the manuscript.

Reviewer #2: The manuscript acknowledges several limitations, including the variability in fNIRS data and the absence of specific control conditions. However, these limitations are not thoroughly discussed in terms of their potential impact on the interpretation of the results.

Additionally, in Figure 1, it is noted that a few channels have a length of less than 30mm, indicating short channels. However, the manuscript fails to provide an explanation for why these short channels are used or why the focus is solely on channels with 30mm-separated optodes. Clarifying the rationale behind the inclusion of short channels or providing justification for focusing on specific channel configurations would enhance the understanding of the experimental setup.

Furthermore, Figures 3, 4, and 5 could be improved in terms of visualization. The lines of the plots are too light in color, making them difficult to distinguish, and the figures appear blurred. Enhancing the contrast of the lines and ensuring better resolution would significantly improve the clarity and interpretability of the figures, thereby enhancing the overall presentation of the results.

6. PLOS authors have the option to publish the peer review history of their article (what does this mean?). If published, this will include your full peer review and any attached files.

Reviewer #1: **Yes: **Hammad Nazeer

Reviewer #2: No

---

## [Author Response · Author response to Decision Letter 0]

19 Apr 2024

Journal Requirements:

> The manuscript cover page and various stylistic changes have been changed to adhere to PLOS ONE's style requirements. These changes were NOT highlighted in the revised manuscript.

2. Please expand the acronym “NIH-NIDCD” (as indicated in your financial disclosure) so that it states the name of your funders in full.

>The requested changes have been made.

3. We note that you have indicated that there are restrictions to data sharing for this study. For studies involving human research participant data or other sensitive data, we encourage authors to share de-identified or anonymized data. Please update your Data Availability statement in the submission form accordingly.

> Unprocessed fNIRS data and speech intelligibility data, along with stimuli used in the study, are now publicly available on Open Science Framework. URL has been included in the manuscript.

> The requested information has been included.

> The requested changes have been made.

Additional Editor Comments:

> These have been addressed separately in the Rebuttal Letter.

---

## [Decision Letter · Decision Letter 1]

14 May 2024

PONE-D-23-42460R1Cortical mechanisms of across-ear speech integration investigated using functional near-infrared spectroscopy (fNIRS)PLOS ONE

Dear Dr. Sobczak,

Thank you for submitting your manuscript to PLOS ONE. After careful consideration, we feel that it has merit but does not fully meet PLOS ONE’s publication criteria as it currently stands. Therefore, we invite you to submit a revised version of the manuscript that addresses the points raised during the review process.

**Minor revisions are still required. **

We look forward to receiving your revised manuscript.

Kind regards,

Noman Naseer, PhD

Academic Editor

PLOS ONE

Journal Requirements:

Additional Editor Comments:

Minor revisions are still required.

Reviewers' comments:

Reviewer's Responses to Questions

**Comments to the Author**

1. If the authors have adequately addressed your comments raised in a previous round of review and you feel that this manuscript is now acceptable for publication, you may indicate that here to bypass the “Comments to the Author” section, enter your conflict of interest statement in the “Confidential to Editor” section, and submit your "Accept" recommendation.

Reviewer #1: (No Response)

Reviewer #2: All comments have been addressed

2. Is the manuscript technically sound, and do the data support the conclusions?

Reviewer #1: Yes

Reviewer #2: Yes

3. Has the statistical analysis been performed appropriately and rigorously? 

Reviewer #1: Yes

Reviewer #2: Yes

4. Have the authors made all data underlying the findings in their manuscript fully available?

Reviewer #1: Yes

Reviewer #2: Yes

5. Is the manuscript presented in an intelligible fashion and written in standard English?

Reviewer #1: Yes

Reviewer #2: Yes

6. Review Comments to the Author

**Reviewer #1:** The aurthors have adequately addressed all my concerns in the previous round of review. However these minor concerns need to be addressed.

1. At multiple places refrence are not cited properly, "Error! Reference source not found" is found especially under the heading, Stimuli (Page 7), page 10, Behavioral results – speech intelligibility experiment (Page 14), and fNIRS responses – DLPFC and AC (Page 15-16). The references and their citatiojn in the text need to be reviewed and corrected thoroughly in the comlete manuscript.

2. In ethical statement, ethical approval number from Institutional Review Board should be mentioned.

**Reviewer #2: **After a thorough re-evaluation of your manuscript and considering the revisions made in response to the previous review, I am pleased to inform you that your paper has met the standards for publication. Your diligent efforts in addressing the previous concerns have substantially improved the quality and clarity of your work.

7. PLOS authors have the option to publish the peer review history of their article (what does this mean?). If published, this will include your full peer review and any attached files.

Reviewer #1: **Yes: **Syed Hammad Nazeer Gilani

Reviewer #2: No

---

## [Author Response · Author response to Decision Letter 1]

2 Jun 2024

I have not included a formal Response to Reviewers as the revisions were minor and solely of a formatting nature, rather than addition of any new content. I have made the requested changes including updating the figure and table links and associated bookmarks so that they are no longer broken in the newest version of the manuscript. In the marked-up manuscript, blue and bolded text remains the same from the previous submission, however I have added the IRB approval number to my Ethical Statement as requested.

---

## [Decision Letter · Decision Letter 2]

2 Jul 2024

Cortical mechanisms of across-ear speech integration investigated using functional near-infrared spectroscopy (fNIRS)

PONE-D-23-42460R2

Dear Dr. Sobczak,

We’re pleased to inform you that your manuscript has been judged scientifically suitable for publication and will be formally accepted for publication once it meets all outstanding technical requirements.

Kind regards,

Noman Naseer, PhD

Academic Editor

PLOS ONE

Additional Editor Comments (optional):

The paper has been revised and can be accepted now.

Reviewers' comments:

Reviewer's Responses to Questions

**Comments to the Author**

1. If the authors have adequately addressed your comments raised in a previous round of review and you feel that this manuscript is now acceptable for publication, you may indicate that here to bypass the “Comments to the Author” section, enter your conflict of interest statement in the “Confidential to Editor” section, and submit your "Accept" recommendation.

Reviewer #1: All comments have been addressed

2. Is the manuscript technically sound, and do the data support the conclusions?

Reviewer #1: Yes

3. Has the statistical analysis been performed appropriately and rigorously? 

Reviewer #1: Yes

4. Have the authors made all data underlying the findings in their manuscript fully available?

Reviewer #1: (No Response)

5. Is the manuscript presented in an intelligible fashion and written in standard English?

Reviewer #1: Yes

6. Review Comments to the Author

Reviewer #1: The atuthors have adequately addressed all my concerns and manuscript is ready for publication in its current form.

7. PLOS authors have the option to publish the peer review history of their article (what does this mean?). If published, this will include your full peer review and any attached files.

Reviewer #1: **Yes: **Syed Hammad Nazeer Gilani

---

## [Editor Report · Acceptance letter]

10 Jul 2024

PONE-D-23-42460R2 

PLOS ONE

Dear Dr. Sobczak, 

I'm pleased to inform you that your manuscript has been deemed suitable for publication in PLOS ONE. Congratulations! Your manuscript is now being handed over to our production team.

Kind regards, 

on behalf of

Dr. Noman Naseer 

Academic Editor

PLOS ONE